# Bayesian Computation Meets Topology

**Julius von Rohrscheidt**                                     *julius.rohrscheidt@helmholtz-munich.de*
*Technical University of Munich, Helmholtz Munich*

**Bastian Rieck**[*]                                          *bastian.grossenbacher@unifr.ch*
*University of Fribourg, Technical University of Munich, Helmholtz Munich*

**Sebastian M. Schmon**[*]                                     *sebastian.schmon@gmail.com*
*Altos Labs, Cambridge, UK*[†]

**Reviewed on OpenReview:** *https://openreview.net/forum?id=Oh1DtRK6dA&noteId=Oh1DtRK6dA*

## Abstract

Computational topology recently started to emerge as a novel paradigm for characterising the 'shape' of high-dimensional data, leading to powerful algorithms in (un)supervised representation learning. While capable of capturing prominent features at multiple scales, topological methods cannot readily be used for Bayesian inference. We develop a novel approach that bridges this gap, making it possible to perform parameter estimation in a Bayesian framework, using topology-based loss functions. Our method affords easy integration into topological machine learning algorithms. We demonstrate its efficacy for parameter estimation in different simulation settings.

## 1 Introduction

Topological machine learning methods enable describing the 'shape' of data at multiple scales while remaining impervious to many different types of noise. This led to strong hybrid models that combine geometry and topology in different domains, including computer vision (Hu et al., 2019; Waibel et al., 2022), graph learning (Horn et al., 2022; Yan et al., 2022; Zhao et al., 2020), time series analysis (Zeng et al., 2021), and representation learning (Carrière et al., 2020; Moor et al., 2020). Despite the advantageous properties of such integrations, topological methods cannot be readily integrated into Bayesian inference frameworks. However, the need to deal with topological information following Bayesian paradigms frequently occurs in real-world data analysis tasks, for example whenever the *shape* of data is informative for the underlying inference task, or in cases where either uncertainty quantification is crucial or available data is not sufficiently large for approaches based on neural networks. In such scenarios, assuming a known prior distribution $p(\theta)$ and real-world data $y$, one could employ Bayes' formula to obtain the *posterior* distribution $p(\theta|y) = {p(y\,|\,\theta)}/{p(y)}p(\theta)$. The posterior distribution offers multiple ways to gain point estimators of the parameter, and allows for a comprehensive description of parameter uncertainties. Regrettably, the application of this formulation to complex generative processes presents substantial challenges, because the likelihood function $p(y\,|\,\theta)$, and consequently, the marginal likelihood $p(y) = \int p(y\,|\,\theta)p(\theta)\mathrm{d}\theta$, frequently become *analytically intractable*. Moreover, the complexity in such systems requires methods that capture structural properties that are invariant to isometries (such as rotations and translations), making computational topology a well-suited tool (Hensel et al., 2021). Topology-driven Bayesian computation needs to satisfy two key requirements, namely (i) it must effectively utilise topological descriptors in the observed data $y$, and (ii) it should offer a theoretically sound notion of posterior beliefs. Both criteria are critical for overcoming the limitations of topological machine learning methods in complex scenarios and enhancing their applicability to real-world data analysis tasks, when the available amount of data is too small to utilise neural networks (e.g. due to

---

[*]These two authors contributed equally as last authors.
[†]work carried out while employed at Durham University.

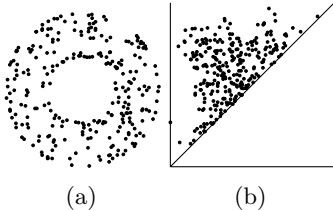

(a)    (b)

Figure 1: Given a point cloud (a), *persistent homology* lets us obtain shape descriptors known as *persistence diagrams* (b). Due to their complex nature, the space of persistence diagrams is not directly amenable to a Bayesian treatment. We overcome this limitation by using *loss functions* that characterise differences between observed and simulated samples.

expensive simulations), or when uncertainty quantification is important. To effectively articulate our posterior beliefs, we describe the scenario of this paper: In the absence of a likelihood function, access to information on the parameter $\theta$ is *solely available* through samples from generative simulation models, which we denote by $x \sim p(x \,|\, \theta)$. Moreover, we do *not* assume $p(x \,|\, \theta)$ to be differentiable in $\theta$, preventing parameter inference via neural networks. This assumption is crucial for a plethora of chaotic models such as collective motion, since such systems often exhibit tipping points that lead to highly non-continuous behaviour. Instead, we assume that we have access to a *loss function* of the form $(x, y) \mapsto \ell(x, y)$, which measures to what extent a generated sample $x$ is 'close' to the observed data $y$. In the general setting as outlined above, we demonstrate that the distribution $\pi(\theta \,|\, y) = \int \exp(-\ell(y,x))p(x \,|\, \theta)p(\theta)\mathrm{d}x / \iint \exp(-\ell(y,x))p(x \,|\, \theta)p(\theta)\mathrm{d}x\mathrm{d}\theta$ offers an alternative for encapsulating the inherent uncertainty in such generative models.

The fundamental **challenge** with this approach is to find a loss function $\ell$ that is suitable for the inference problem at hand. While traditional approximate Bayesian computation (ABC) often makes use of losses based on summary statistics derived from the given data, such statistics are not straightforward to define, and usually far from being sufficient. By contrast, geometrical losses like the Hausdorff distance often do not capture the necessary information in order to perform accurate inference.

As the **central contributions** of this paper, we (i) leverage Bayesian inference and topology via efficient geometrical–topological loss terms that are capable of handling different data modalities, (ii) show that the comparison-based posterior, which we define in Section 3, can be seen as the solution to a generalised Bayesian optimization problem, (iii) demonstrate how sampling algorithms can leverage topologically-informed loss functions to perform Bayesian inference, and (iv) we demonstrate the utility of our topology-aware approach on several complex generative models. Since our approach is Bayesian by nature, prior knowledge about the parameters to be inferred can readily be integrated into the workflow, making our workflow interpretable in comparison to black-box models.

> **In a nutshell:** Our framework fits into the realm of simulation-based inference, where an exact likelihood function may not be determined, but with a simulating model being accessible in order to sample data from the likelihood. The main novelty consists of incorporating topological information into parameter inference and showing its utility for parameter estimation in complex systems.

## 2   Background: Computational Topology

Before we introduce our approach, we first provide a brief overview of relevant methods from computational topology. Such methods recently emerged, making use of geometrical–topological properties of data to describe their overall 'shape.' The concept of *persistent homology* is of particular interest in the context of machine learning since it leads to efficient descriptors of structured and unstructured data while at the same time satisfying invariance and robustness guarantees (Hensel et al., 2021).

Persistent homology obtains multi-scale geometrical–topological shape descriptors of data by means of combinatorial data structures. Given a metric such as the Euclidean distance, *persistent homology* assigns a family of shape descriptors, the *persistence diagrams*, to point clouds (Edelsbrunner & Harer, 2010);

the process can be extended to other modalities. The calculation of persistence diagrams turns out to be stable with respect to the Hausdorff distance between point clouds, making this method highly robust in practice (Cohen-Steiner et al., 2007). See Fig. 1 for an overview of this process.

The main idea is to obtain a nested sequence of simplicial complexes from the given data that encodes topological information about the data *on multiple scales*. The notion of 'scale' and how to define such a sequence in practice depends on the given problem; we will describe two constructions that are relevant in the context of this paper. Given a point cloud $\mathsf{X}$, the *Vietoris-Rips complex* at stage $\epsilon$ associated to $\mathsf{X}$ is the abstract simplicial complex in which a $k$-simplex is defined for every subset of $\mathsf{X}$, consisting of $k + 1$ points in $\mathsf{X}$ that have diameter at most $2\epsilon$. By varying $\epsilon$ one obtains a sequence of nested simplicial complexes associated to $\mathsf{X}$. The second construction is given by *cubical complexes*. Since we only use cubical complexes for greyscale images, we restrict ourselves to this particular setting, and refer to the literature for a more comprehensive introduction (Rieck et al., 2020b; Wagner et al., 2012). Any greyscale image $A$ can be interpreted as a graph, where the nodes are given by the non-zero pixels, such that there is an edge between two nodes if and only if the two nodes correspond to neighbouring pixels. For a threshold $\epsilon$, we obtain a new image from $A$ by setting all pixels in $A$ to zero, which admit a pixel value less than $\epsilon$. Varying $\epsilon$ leads to a nested sequence of the graphs corresponding to the respective images, and therefore we obtain a nested sequence corresponding to $A$. Subsequently, we explain how to derive *multi-scale topological features* from such a nested sequence, also known as a *filtration*:[1] Let $\emptyset = \mathrm{K}_0 \subseteq \mathrm{K}_1 \subseteq \cdots \subseteq \mathrm{K}_{m-1} \subseteq \mathrm{K}_m = \mathrm{K}$ be a nested sequence of simplicial or cubical complexes. The main idea of persistent homology involves tracking topological features, measured by means of *homology groups*, alongside this filtration. The family of boundary operators $\partial(\cdot)$, together with the inclusion homomorphism, induces a homomorphism between corresponding *homology groups* of the filtration, i.e. $\iota_d^{i,j} : H_d(\mathrm{K}_i) \to H_d(\mathrm{K}_j)$. [2] This mapping yields a sequence of homology groups for every dimension $d$. Given indices $i \leq j$, the $d$th *persistent homology group* is then defined as $H_d^{i,j} := \ker \partial_d(\mathrm{K}_i) / (\operatorname{im} \partial_{d+1}(\mathrm{K}_j) \cap \ker \partial_d(\mathrm{K}_i))$. It can be seen as the homology group that contains all homology classes created in $\mathrm{K}_i$ that are still present in $\mathrm{K}_j$. Typically, the filtration of $\mathrm{K}$ has associated values $a_0 \leq a_1 \leq \cdots \leq a_{m-1} \leq a_m$ (such as edge weights of a graph or pairwise distances of a point cloud). This permits us to summarise multi-scale topological activity in a *persistence diagram*: for each dimension $d$ and each pair $i \leq j$, we store the pair $(a_i, a_j) \in \mathbb{R}^2$ with multiplicity $\mu_{i,j}^{(d)} := \left( \beta_d^{i,j-1} - \beta_d^{i,j} \right) - \left( \beta_d^{i-1,j-1} - \beta_d^{i-1,j} \right)$ in a set (in practice, $\mu_{i,j}^{(d)} = 0$ for many pairs). The resulting set of points is called the $d$th *persistence diagram* $\mathcal{D}_d$. Given a point $(x, y) \in \mathcal{D}_d$, the quantity $\operatorname{pers}(x, y) := |y - x|$ is referred to as its *persistence*. High persistence is commonly considered to correspond to *features*, while low persistence is seen to indicate *noise* Edelsbrunner & Harer (2010), but recent work shows that in many data sets, low persistence values rather indicate 'low reliability,' which may still play an important role for downstream analyses (Bendich et al., 2016; Rieck et al., 2020b).

## 3 Methods

### 3.1 Bayesian Inference

Within the Bayesian framework, unknown quantities, such as parameters denoted by $\theta$, are furnished with a probability distribution $p(\theta)$, referred to as the *prior distribution*. Subsequent inference about the parameters $\theta$ given observed data $y$ is performed via the Bayesian update procedure. This process involves adjusting the prior distribution $p(\theta)$ through the multiplication by the ratio of the likelihood to the evidence, i.e. $p(\theta|y) = {p(y|\theta)}/{p(y)}p(\theta)$. However, applying this procedure in our particular context presents notable challenges. The primary issue stems from the fact that the likelihood function $p(y\,|\,\theta)$ is typically *unknown* or *analytically intractable*, thus posing considerable difficulties for estimation, particularly when the data exhibits intricate geometrical–topological properties. To circumvent this problem, recent research has shown that the use of the likelihood function of the data can be avoided, producing so-called generalised Bayesian posteriors (Bissiri et al., 2016). One approach to construct such a generalisation to the Bayesian posterior can be found through the lens of viewing Bayesian inference as the solution to an optimisation problem (Csiszár, 1975; Donsker

---

[1] For a more in-depth discussion of computational topology, in particular in the context of machine learning, we refer the reader to Hensel et al. (2021).

[2] A high-level explanation of homology groups is contained in the appendix.

& Varadhan, 1975; Zellner, 1988). Specifically, it can be seen as finding a distribution $q$ that balances prior information as measured by the Kullback–Leibler (KL) divergence between the prior and the posterior candidate $q$ (which we want to *minimise*) and the expected log-likelihood of new observations $y$ (which we want to *maximise*):

$$\arg\min_{q\in\mathcal{P}(\Theta)}\left\{\mathbb{E}_q[-\log p(y|\theta)] + \mathrm{KL}(q(\theta), p(\theta))\right\}$$

The solution to this problem is the traditional Bayesian posterior $q^*(\theta) = p(\theta|y)$. Recently, generalisations of this approach have been suggested, demonstrating that coherent belief updates can also occur if instead of the negative log-likelihood loss, $-\log p(y\,|\,\theta)$ any loss $l(y,\theta)$ is taken (Bissiri et al., 2016; Knoblauch et al., 2022). The solution to the concomitant optimisation problem

$$\arg\min_{q\in\mathcal{P}(\Theta)}\left\{\mathbb{E}_q[l(y,\theta)] + \mathrm{KL}(q(\theta), p(\theta))\right\} \tag{1}$$

is given by

$$q^*(\theta) = \frac{\exp(-l(y,\theta))p(\theta)}{\int \exp(-l(y,\theta))p(\theta)\mathrm{d}\theta} =: \pi_{\mathrm{GBI}}(\theta\,|\,y). \tag{2}$$

However, the problem is that even in this general formulation, the loss formally scores observed data $y$ with respect to a parameter $\theta$. A common approach is to use expectations over samples from a generative model, i.e. simulate $x \sim f(x\,|\,\theta)$ and approximate

$$l(y,\theta) := E_{p(x\,|\,\theta)}\left[\ell(y,x)\right] \approx \frac{1}{N}\sum_{i=1}^{N}\ell(y,x_i), \tag{3}$$

with $x_i \sim p(x\,|\,\theta)$. Replacing the loss $l(y,\theta) = E_{p(x\,|\,\theta)}\left[\ell(y,x)\right]$ with a sample average, however, changes[3] the posterior from Eq. (2). We will demonstrate in the following that we can directly construct a valid posterior from the loss function $\ell(y,x)$.

## 3.2 Comparison-Based Posteriors

Uncertainty estimates predicated on distance functions have a long-standing history in Bayesian statistics as a means of *approximating* posterior distributions, with notable use in the realm of *approximate Bayesian computation* (ABC) (Tavaré et al., 1997; Pritchard et al., 1999; Beaumont et al., 2002). In the ensuing discussion, we will illustrate that a modification of the aforementioned approach, specifically, a generalised posterior from ABC such as Schmon et al. (2021), can furnish us with a theoretically sound form of posterior update. The proposed approach involves using a *loss function* $(x,y) \mapsto \ell(y,x)$ to construct a joint posterior involving simulated data $x \sim p(x\,|\,\theta)$ and real data $y$. It is of the form

$$\pi(\theta, x\,|\,y) \propto \exp\left(-\ell(y,x)\right)p(x\,|\,\theta)p(\theta). \tag{4}$$

This approach, however, encounters several obstacles. Firstly, it is not clear that Eq. (4) defines a reasonable notion of uncertainty. In addition, since $x$ is just *simulated* data, not *observed* data, we need to average over possible values, leading to

$$\pi(\theta\,|\,y) = \frac{\int_{\mathsf{X}} \exp\left(-\ell(y,x)\right)p(x\,|\,\theta)p(\theta)\mathrm{d}x}{\iint_{\mathsf{X}\times\Theta} \exp\left(-\ell(y,x)\right)p(x\,|\,\theta)p(\theta)\mathrm{d}x\mathrm{d}\theta}. \tag{5}$$

It turns out that this is is a valid (generalised) posterior under the loss associated with the topological properties of our data, namely:

**Proposition 3.1.** *The comparison-based posterior*

$$\pi(\theta, x\,|\,y) = \frac{\exp\left(-\ell(y,x)\right)p(x\,|\,\theta)p(\theta)}{\int_{\mathsf{X}}\int_{\Theta} \exp\left(-\ell(y,x)\right)p(x\,|\,\theta)p(\theta)\mathrm{d}\theta\mathrm{d}x} \tag{6}$$

---

[3]This occurs even in the case where the estimator is unbiased since this property is lost under the non-linear (exponential) transformation by virtue of Jensen's inequality.

*is the solution $q^*(\theta, x) = \pi(\theta, x \mid y)$ to the optimisation problem*

$$\arg\min\left\{\mathbb{E}_q[\ell(y, x)] + \text{KL}\left(q(\theta, x), p(x|\theta)p(\theta)\right)\right\}, \tag{7}$$

*where the $\arg\min$ ranges over all $q \in \mathcal{P}(\Theta \times \mathsf{X})$.*

The proof can be found in the appendix. In Eq. (7), the first term in the objective function measures the expected loss under the posterior, where the loss function $\ell(y, x)$ quantifies the discrepancy between the observed data $y$ and the simulated data $x$. This is precisely where we can use improved *inductive biases* or domain knowledge and employ loss functions such as the topological losses introduced in Section 3.4. The second term encourages the posterior to stay close to the prior, as in the traditional Bayesian setting, however, with a significant difference: the prior distribution $p(\theta)$ over the parameters $\theta$ together with the simulation model $p(x \mid \theta)$ naturally implies prior beliefs over what the data should look like. Thus, it is sensible to consider the KL-divergence between the joint beliefs $p(x \mid \theta)p(\theta)$ and $q(x, \theta)$.

Proposition 3.1 shows that the comparison-based posterior of Eq. (4) is in fact a solution to the respective optimisation problem (which is the analogue of Eq. (1) in our setting). The motivation is therefore to use comparison-based posteriors for approximate inference under the assumption that an appropriate loss function is available. Proposition 3.1 does not necessitate a topological loss function and can be considered independently of a specific loss function. However, in particular, it permits incorporating topological information via topological losses.

To the best of our knowledge, the argument and proof that comparison-based posteriors can be seen as the solution to a generalised Bayesian optimization problem is new. While distance-based loss functions, where a parameter $\theta$ is only available through simulation have been used before, they are commonly considered approximations of an idealised loss function

$$\ell(y, \theta) := \lim_{n \to \infty} \sum_{i=1}^{n} \ell(y, x_i). \tag{8}$$

Thus, Proposition 3.1 demonstrates that comparison-based posteriors do not have to be seen as a 'suboptimal' approximation to an abstract ideal, but can be seen as a valid generalised posterior in their own right.

As such, our approach can be seen as a generalisation of the traditional Bayesian posterior in the sense that it allows for a more flexible modelling of the data generation process. Instead of assuming that the observed data is generated exactly according to the model $p(y \mid \theta)$, we allow for the possibility that the data is generated from a distribution that produces data *close* to $y$ in terms of the loss function $\ell$. This makes our approach potentially more robust to model misspecification and more suitable for complex data with intricate topological properties.

### 3.3 Inference for Comparison-Based Posteriors

Evaluating posteriors such as those given by Eq. (5) analytically is typically *infeasible*. An alternative approach may involve direct targeting of a variational distribution as shown in Eq. (7). However, this strategy encounters two main difficulties: firstly, the distribution $p(x \mid \theta)$ is often intractable, rendering the computation of the KL divergence term non-trivial. Secondly, in order to compute the gradients, we must propagate them through the simulation models. This poses a challenge as $p(x \mid \theta)$ often represents a 'black-box' model, not necessarily implemented within a framework that supports automatic differentiation. In fact, many complex simulation models are not even differentiable in the parameters to be optimised. Additionally, such algorithms typically require large amounts of data, which is a frequent limitation when simulations are computationally expensive. Consequently, this typically precludes the practical application of gradient-based optimisation techniques. However, it turns out that Monte Carlo algorithms, such as self-normalised importance sampling or so-called pseudo-marginal Markov chain Monte Carlo (Andrieu & Roberts, 2009) are able to compute expectations with respect to posteriors of the form in Eq. (5).

---

**Algorithm 1** Importance sampling estimation of the mean

---

1: **Input:** Observed data $y$, test function $h$
2: **for** $i = 1 : n$ **do**
3:     Sample $\theta_i \sim p(\theta)$
4:     Sample $x_i \sim p(x \,|\, \theta_i)$
5: **end for**
6: **Output:**

$$\hat{h}(\theta) = \frac{\sum_{i=1}^n h(\theta_i) e^{-w\ell(y,x_i)}}{\sum_{i=1}^n e^{-w\ell(y,x_i)}}$$

---

Algorithm 1 shows a self-normalised *importance sampling* procedure. Almost-sure convergence can be shown under mild regularity conditions (e.g. Owen, 2013, Theorem 9.2) for a test function of interest $h$, i.e.

$$\frac{\sum_{i=1}^n h(\theta_i) e^{-\ell(y,x_i)}}{\sum_{i=1}^n e^{-\ell(y,x_i)}} \tag{9}$$

approaches

$$\frac{\iint h(\theta) e^{-\ell(y,x)} p(x \,|\, \theta) p(\theta) \mathrm{d}x \mathrm{d}\theta}{\iint e^{-\ell(y,x)} p(x \,|\, \theta) p(\theta) \mathrm{d}x \mathrm{d}\theta} = E_{\pi(\theta \,|\, y)}[h(\boldsymbol{\theta})] \tag{10}$$

as $n \to \infty$. An alternative is to use *Markov chain Monte Carlo* (MCMC), where we produce a Markov chain converging to the distribution (5); see Algorithm 2 in the appendix. Such algorithms indeed converge to the desired target distribution (Andrieu & Roberts, 2009, Theorem 1) under mild assumptions. A high-level overview of pseudo-marginal MCMC is contained in the appendix.

## 3.4 Losses Based on Geometry & Topology

Having introduced multiple inference algorithms and explained how to obtain topological features, we now discuss how to derive topology-based loss functions. We start with discussing metrics in computational topology. It turns out that persistence diagrams can be endowed with a metric by using optimal transport. Given two diagrams $\mathcal{D}$ and $\mathcal{D}'$ containing features of the same dimensionality, their $p$th *Wasserstein distance* is defined as

$$\mathrm{W}_p(\mathcal{D}, \mathcal{D}') := \left( \inf_{\eta \colon \mathcal{D} \to \mathcal{D}'} \sum_{x \in \mathcal{D}} \|x - \eta(x)\|_\infty^p \right)^{\frac{1}{p}}, \tag{11}$$

where $\eta(\cdot)$ denotes a bijection. Since $\mathcal{D}$ and $\mathcal{D}'$ generally have different cardinalities, we consider them to contain an infinite number of points of the form $(\tau, \tau)$, i.e. tuples of zero persistence; this is akin to requiring each diagram to contain the projections of points to the diagonal, originating from the *other* diagram. A suitable $\eta(\cdot)$ can thus always be found. Solving Eq. (11) is practically feasible using modern optimal transport algorithms (Flamary et al., 2021). While this is the most 'principled' approach—in the sense that such a loss formulation forms a proper metric in the mathematical sense—alternative formulations to Eq. (11) exist, and our framework is fundamentally compatible with all of them. Along these lines, other topological descriptors, such as *Betti curves* (Rieck et al., 2020a) or *persistence images* (Adams et al., 2017), might be used instead of persistence diagrams. These descriptors are often easier to compute *and* afford the use of fast $L_1$ or $L_2$ distances as proxies for Eq. (11). For instance, the $L_2$ distance between persistence images is known to share some advantageous properties with the Wasserstein distance (Adams et al., 2017) but it only handles a 'discretised' version of the data, so some information is invariably lost (or, to briefly consider an optimistic point of view, the persistence image is *smoother* than a persistence diagram). As another alternative to the previously-described descriptors, we could also employ a *kernel*, i.e. a similarity measure between persistence diagrams. These similarity measures are not metrics in the mathematical sense, lacking the requirement of the identity of indiscernibles, but are easier to compute (unlike the persistence images, they do not require vectorisations) and require fewer parameters (Kwitt et al., 2015; Reininghaus et al., 2015).

**A loss function based on the Wasserstein distance.** To leverage the power of computational topology and Bayesian inference, we propose combining the Bayesian framework for updating posterior probability estimates with a *topological loss function*. Given observed data $y$ and simulated data $x$ sampled from $p(x \,|\, \theta)$, let $\mathcal{D}_x$ and $\mathcal{D}_y$ be the persistence diagrams corresponding to $x$ and $y$, respectively. We then define a *topological loss function* $\ell_T \colon \mathsf{Y} \times \mathsf{X} \to \mathbb{R}$, by setting

$$\ell_T(y, x) := \mathrm{W}_p(\mathcal{D}_y, \mathcal{D}_x). \tag{12}$$

This formulation applies to different data modalities. If the data can be represented as a point cloud, we calculate persistence diagrams from the Vietoris–Rips complex. For (greyscale) image data, we obtain persistence diagrams via *cubical complexes*, enabling topological feature calculations for more complicated modalities such as MRI data Rieck et al. (2020b). As we outlined above, other choices for the loss function would be possible, and we may now continue with our Bayesian updating procedure. Notice that unless otherwise mentioned, we use $p = 2$ to compute the topological loss.

Moreover, we note that the implementation of `giotto-tda` (Tauzin et al., 2020), which we use in our experiments, uses a faster approximate algorithm rather than the actual loss. Under the hood, this implementation uses an auction algorithm with a worst-case computational complexity of $\mathcal{O}\big(n \cdot m \cdot \log\big(\epsilon^{-1}\big)\big)$, with $n$ and $m$ being the number of points in the respective persistence diagrams and $\epsilon$ being a precision parameter.

**Properties.** The Wasserstein distance between persistence diagrams is *stable* in the sense that the geometric distance is an upper bound of the topological distance. Given two point clouds $\mathsf{Y}, \mathsf{X}$ of the same cardinality, we have

$$\mathrm{W}_p(\mathcal{D}_y, \mathcal{D}_x) \leq C_p \, \mathrm{W}_p(\mathsf{Y}, \mathsf{X}), \tag{13}$$

where $C_p$ refers to a constant that only depends on $p$ and $\mathrm{W}_p(\mathsf{Y}, \mathsf{X})$ is defined similarly to Eq. (11), but calculated on the *point clouds themselves* Skraba & Turner (2022). Since $\mathrm{W}_p(\cdot, \cdot) \leq \mathrm{W}_{p'}(\cdot, \cdot)$ for $p' \leq p$, low values of $p$ are desirable in terms of stronger stability; we find that $p = 2$ provides a suitable compromise solution. Moreover, our loss function $\ell_T$ is invariant under isometries (Edelsbrunner & Harer, 2010) and stable under subsampling (Moor et al., 2020). Eq. (13) can be interpreted as follows: once $\mathsf{X}$ and $\mathsf{Y}$ are geometrically close, their corresponding topological properties are similar as well. However, the converse is *not true* in general since structural similarity with respect to topological properties may occur between two datasets although they are *not* close geometrically. In fact, the left-hand side of Eq. (13) is often substantially smaller than its right-hand side, which serves as the initial motivation to use topological losses instead of their geometrical counterparts, particularly for inference tasks in which the shape of data is informative for the underlying objective. We validate this theoretical observation empirically in Section 5, by showing that topology encoded in losses indeed lead to more accurate parameter inference.

**A loss function based on the Hausdorff distance.** As a baseline and computationally simpler comparison partner, we also define a geometry-based loss function based on the *Hausdorff distance* between point clouds $\mathsf{X}$ and $\mathsf{Y}$. Given a metric space $(M, d)$ and two non-empty subsets $\mathsf{X}, \mathsf{Y} \subseteq M$, we define our loss as

$$\ell_G(\mathsf{X}, \mathsf{Y}) := \inf\{\epsilon \geq 0 \,|\, \mathsf{X} \subset \mathsf{Y}_\epsilon, \mathsf{Y} \subset \mathsf{Y}_\epsilon\}, \tag{14}$$

where $\mathsf{X}_\epsilon = \cup_{x \in \mathsf{X}}\{m \in M; \ d(m, x) \leq \epsilon\}$ denotes the $\epsilon$-thickening of $\mathsf{X}$ in $M$. While this loss does not satisfy invariance properties, it is more efficient to compute in practice. However, as we will see in the experimental section, its utility in complex data generation scenarios is limited. For $|X| = n$ and $|Y| = m$, the computational complexity of the Hausdorff distance is $\mathcal{O}(n \cdot m)$.

## 4 Related Work

A range of Bayesian inference methods have been developed to address the challenges involved with uncertainty quantification when dealing with complex models whose likelihoods are analytically intractable. While each of these approaches offers a distinct solution, they also present their own unique sets of challenges and limitations.

Approximate Bayesian computation (ABC) is a well-established approach (Tavaré et al., 1997; Pritchard et al., 1999; Beaumont et al., 2002). ABC employs a pre-defined threshold to accept parameters that yield a small enough distance between the simulated and observed data. More recently, Wasserstein ABC (Bernton et al., 2019) has capitalised on optimal transport theory to define a distance function between simulated and observed data. This form of rejection-based ABC comes with several potential problems such as the choice of the rejection threshold: a small threshold will lead to high numerical costs whereas a threshold too large can result in incorrect coverage of credible regions (Frazier et al., 2018). To make matters worse, rejection ABC algorithms are known to exhibit non-standard asymptotic behaviour in case of model misspecification (Frazier et al., 2019).

Several authors have suggested constructing a generalised posterior along the lines of Eq. (2) using distance functions, including *maximum mean discrepancy* (MMD) (Park et al., 2015; Chérief-Abdellatif & Alquier, 2020). However, these methods, in contrast to our approach, use a theoretical loss, for instance, $l(y, \theta) = E_{p(x|\theta)}[\ell(y, x)]$, which is then *approximated* using empirical averages, thus potentially introducing a source of bias or instability. By contrast, our approach, is more related to (particle) filtering (e.g. Doucet et al., 2009), where inference is carried out over the latent state simultaneously. Miller & Dunson (2018) introduced a discrepancy-based Bayesian procedure they termed 'coarsening,' where the observed data is considered as a coarsened version of latent data, thereby enhancing robustness.

Finally, topological data analysis (TDA) has already started to see some use in the context of analysing generative models. Some works are concerned with fitting the parameters of previously-defined prior distributions of persistence diagrams (Maroulas et al., 2022; Oballe, 2020), thus demonstrating the expressivity of TDA for *describing* complex systems. We are interested in parameter estimates of black-box models by exploiting this expressivity. Along these lines, Topaz et al. (2015) use persistent homology for a qualitative analysis of the topological characteristics of swarming models (such as Vicsek model). In comparison, our approach is intended to leverage topology and Bayesian inference for *quantitative* studies, enabling parameter estimation. More recently, Thorne et al. (2022), used topological statistics in combination with *rejection sampling.* This method constitutes the first use of TDA in the setting of ABC; it involves numerous parameter choices that our method does *not* necessitate. Moreover, the rejection sampling approach is itself parameter-driven and finding a suitable rejection threshold requires additional domain knowledge. Our approach stands out by integrating a loss function to construct a posterior distribution directly, without resorting to approximation, thereby offering a theoretically-grounded form of posterior belief. In comparison to the aforementioned method, ours does not necessitate the choice of topological summary statistics but operates on the full persistence information.

## 5 Experiments

We empirically validate the utility of our proposed method in the form of three main experiments that compare topological losses and respective geometrical losses, by using the results from Section 3. In the first two experiments, we focus on parameter inference for chaotic simulation models, which can be seen as (approximately) solving an inverse problem. Notice that these are well-suited scenarios for applying topology-driven Bayesian methods, since chaotic models are known to admit likelihood functions that are *neither analytically tractable* (due to their complexity) *nor differentiable* (due to the presence of tipping points, for instance), thus precluding the use of both neural networks and traditional approaches to Bayesian inference. Moreover, our method requires only a very small number of simulations (here, 250) to perform reasonably, making our approach superior to neural networks with regards to the required number of data points (we find that this is particularly relevant in setting where the simulation is (computationally) expensive). Furthermore, for each run of the experiments, we use 100 observation points for inference, and normal distributions for both the prior and the proposal distribution in the MCMC procedure. The weight parameter $w$ in the importance sampling procedure is set to 10. The third experiment involves a synthetic model that shows how our method may be used for inference in the context of imaging data. In all of our experiments, topological losses perform significantly better than a respective geometrical loss. Finally, we compare our results to standard summary statistics that are frequently used in the context of approximate Bayesian computation, see Appendix A. Remarkably, our topology-driven approach is observed to perform significantly better than $L_p$-norms on summary statistics for the first two non-synthetic experiments, underpinning our proposed method's utility

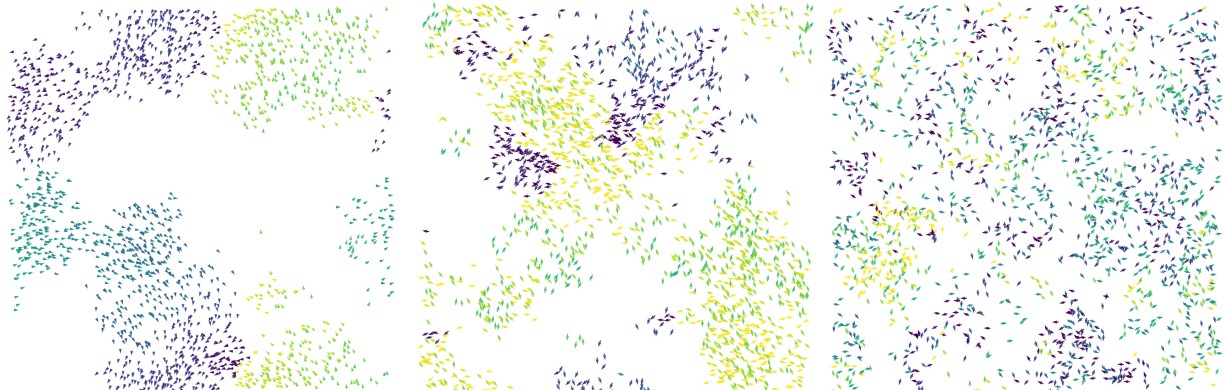

Figure 2: Left to right: simulations of the Vicsek model for $\eta = 0.15, 0.3$ and 0.6, respectively. For lower values of $\eta$ swarming behaviour emerges, whereas for higher values of $\eta$ swarms tend to merge and the model becomes more ergodic.

for practitioners. As a consistency check, we use our proposed methods for inferring the radius parameters of two objects, namely the 2-sphere and the 2-torus. Table 3 in the appendix depicts the results of the inference process.

### 5.1 Vicsek Model: Swarm Behaviour

The Vicsek model is an agent-based model that is particularly useful to study collective motion like swarming and flocking. The underlying idea is that each agent aligns at any time step with its nearest neighbours, before the alignment is perturbed by an additional random term. The resulting angle of the alignment determines the direction of movement for the subsequent time step, where the movement takes place with constant speed which is independent of time and individual. Following Vicsek et al. (1995), the formula for the alignment angle update of an individual $i$ is given by $\theta(t+1) = \langle \theta(t) \rangle_r + \Delta\theta$, where $\langle \theta(t) \rangle_r$ denotes the average of the angles of individuals with distance to agent $i$ at most $r$, and $\Delta\theta$ is uniformly sampled from the interval $[-\eta/2, \eta/2]$ for $0 \leq \eta \leq 2\pi$. The new position of the agent is then determined by moving with constant speed towards the direction of the updated angle. Notice that although the position of each agent can be expressed analytically, the dependencies between agents leads to highly complex behaviour of the overall system over time, which cannot be expressed analytically. We therefore are in the proposed setting of an intractable likelihood, necessitating a simulation-based approach to inference.

To demonstrate the utility of topological losses for parameter inference in a setting where the respective likelihood function is *not analytically tractable*, we infer the noise parameter $\eta$ after a certain number of iterations (time steps) of the model. We distribute 2000 agents across a square of edge length $L = 25$, where opposite edges of the square are identified with the orientation being preserved; the 'world' thus constitutes the surface of 2-torus. For the inference of $\eta$, we use 250 simulations for each sampling method, aiming to infer $\eta$ after $5, 10$, and $50$ iterations (time steps) of the model, respectively, repeating the inference 5 times. Since the Vicsek model gives rise to point clouds, we calculate Vietoris–Rips complexes and compare the corresponding persistence diagrams. As a baseline, we use the Hausdorff distance between the point clouds; Fig. 3 depicts the results. In our experiments, the results for the topological loss with the MCMC sampling procedure outperforms in all of the settings, and leading to highly accurate estimates even after 50 time steps. By contrast, we note that the estimators that were calculated with the Hausdorff loss lead to particularly poor results in the regimes that are closer to being ergodic, i.e. for larger values of $\eta$. Finally, we remark that our approach with the proposed topological loss outperforms standard rejection sampling in combination with summary statistics; see Appendix A.

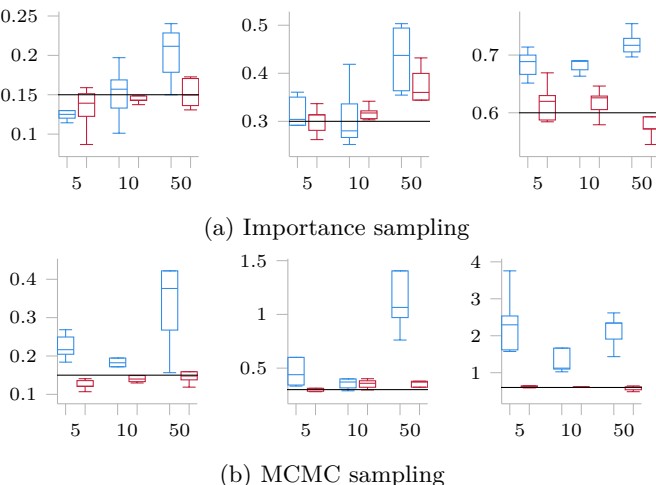

(a) Importance sampling

(b) MCMC sampling

Figure 3: Summary of parameter estimates for the noise parameter $\eta$ of the Vicsek model, using 5 repetitions of the inference procedure. Each subplot depicts estimates for $\eta \in \{0.15, 0.30, 0.60\}$, respectively. The true parameter is shown as a solid black line. The horizontal axis shows the number of iterations of the Vicsek model. Estimates based on the geometrical loss $\ell_G$ are less accurate than the topological loss $\ell_T$, in particular for larger number of iterations. Please refer to Table 4 for the raw values.

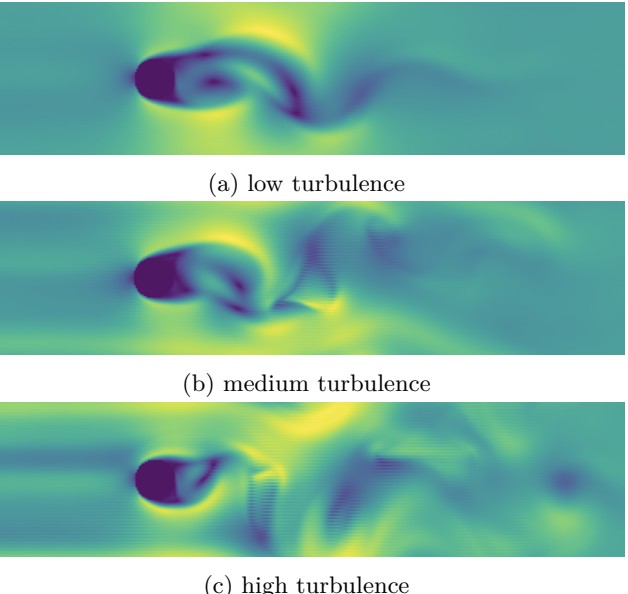

(a) low turbulence

(b) medium turbulence

(c) high turbulence

Figure 4: Left to right: simulations of the fluid model after 3000 iterations, for $\eta = 0.2, 0.3$ and $0.4$, respectively. For lower values of $\eta$, the fluid flow is less turbulent than for higher values of $\eta$.

## 5.2 Fluid Dynamics: Lattice Boltzmann Simulation

The lattice Boltzmann methods (LBM) are a collection of algorithms that are used for fluid simulation. Instead of the conservation of macroscopic properties of the dynamical system, LBM models the fluid over a discrete lattice, by performing local propagation and collision processes, for all particles simultaneously. For a thorough introduction, see Krüger et al. (2017). After initialising the state of each particle equally, a portion of normally distributed noise is added to each state, independently. This portion of randomness is controlled by a global parameter $\eta \geq 0$, where higher values of $\eta$ correspond to a higher portion of noise in the initialisation. The crucial observation is that the degree of turbulence in the system can be modelled by

Table 1: Estimating the randomness parameter of the lattice Boltzmann model after 3000 simulation iterations, using 5 repetitions of the inference procedure. The topological loss together with importance sampling outperforms all of the other configurations.

| | | Randomness parameter $\eta$ | | |
|---|---|---|---|---|
| *Loss* | *Sampling method* | 0.20 | 0.30 | 0.40 |
| $\ell_G$ | Importance sampling | $0.16 \pm 0.02$ | $0.22 \pm 0.04$ | $0.35 \pm 0.13$ |
| | MCMC | $0.15 \pm 0.10$ | $0.91 \pm 1.64$ | $0.38 \pm 0.25$ |
| $\ell_T$ | **Importance sampling** | $\mathbf{0.21 \pm 0.01}$ | $\mathbf{0.27 \pm 0.00}$ | $\mathbf{0.38 \pm 0.01}$ |
| | MCMC | $2.28 \pm 1.40$ | $1.71 \pm 0.81$ | $2.35 \pm 1.10$ |

Table 2: Estimating the parameter of the Percolation model, using 5 repetitions of the inference procedure. The topological loss together with either importance sampling or MCMC sampling outperforms all other configurations.

| | | Randomness parameter $p$ | | |
|---|---|---|---|---|
| *Loss* | *Sampling method* | 0.15 | 0.30 | 0.60 |
| $\ell_G$ | Importance sampling | $0.18 \pm 0.00$ | $0.30 \pm 0.47$ | $0.59 \pm 0.00$ |
| | MCMC | $3.18 \pm 2.55$ | $1.30 \pm 0.47$ | $2.18 \pm 1.47$ |
| $\ell_T$ | **Importance sampling** | $\mathbf{0.15 \pm 0.00}$ | $\mathbf{0.29 \pm 0.00}$ | $\mathbf{0.59 \pm 0.01}$ |
| | **MCMC** | $\mathbf{0.15 \pm 0.02}$ | $\mathbf{0.29 \pm 0.00}$ | $\mathbf{0.59 \pm 0.00}$ |

varying $\eta$, as is shown in Fig. 4. Our objective is therefore to infer the parameter $\eta$, which corresponds to a given observation after a certain amount of iterations in the model, where we fix this number of iterations to be $t = 3000$ time steps. As the topological loss, we use the Wasserstein distance between the persistence diagrams of the *cubical complexes* corresponding to the observed and the simulated state, respectively, while the Hausdorff distance between the states serves as a baseline comparison. Table 1 shows the results; we observe that the topological loss together with importance sampling outperforms the geometrical loss in both configurations, by far. Again, rejection sampling in combination with summary statistics performs less reliably; see Appendix A.

## 5.3 Percolation Model: Multi-Scale Structures

In statistical physics, percolation refers to the behaviour of a network when links are added to it. Percolation can be viewed as a process of geometric phase transition since at a critical state of the network, disconnected components merge into large connected clusters by the addition of a small fraction of links. In our setting, the percolation model is probabilistic in the sense that links are added randomly, with some probability $p$, and the goal is to infer $p$. We realise such a network as a square 2D greyscale image of pixel size $n^2$, for some positive integer $n$.[4] We assign a value $v$ to each pixel in the image, where $v = 0$ with probability $1 - p$, and $v$ is sampled uniformly from the set $\{1, \ldots, v_{\max}\}$ with probability $p$. Here, $v_{\max}$ is the maximum realisable greyscale value, which we set to 50 for our experiments. For the inference of $p$, we used 250 simulations for both the importance sampling and the MCMC sampling. Since we are dealing with greyscale images, we obtain persistence diagrams via *cubical complexes*.

We compare our proposed topological loss $\ell_T$ to several other losses that are commonly used in imaging processing, including MSE, RMSE, the universal image quality index (UQI), the relative average spectral error (RASE), the spatial correlation coefficient (SCC), and the pixel-based visual information fidelity (VIFP). Since we find that the results of SCC and VIFP outperform other scores, and since SCC and VIFP lead to

---

[4]Note that alternatively, a greyscale image can be interpreted as an undirected graph, by defining the set of vertices to be the set of pixels, and adding edges between neighbouring pixels if both of their values are non-zero.

comparable results, we only show the results for SCC (see Zhou et al. (1998) for a construction of SCC) in comparison to the ones obtained by the topological loss. Table 2 shows the results. We again observe that our topology-based loss $\ell_T$, together with MCMC sampling, outperforms the other methods by far.[5]

However, in this particular experiment, rejection sampling together with a mean statistic performs even better. This is due to the idiosyncratic behaviour of the percolation model, which enables inferring the true parameter from the *mean* of pixel values; see Appendix A for an extended discussion. In *all* of our experiments, we observe that topological losses outperform a respective geometrical loss. This observation highlights the utility of topology in simulation-based inference: In particular complex systems exhibit typical collective behaviour, which can be captured by their topological descriptors (i.e. their 'shape'), and which cannot be captured via geometrical losses, since geometric properties of the output of a simulating model (such as the *exact* positions of individual agents) may differ even for a fixed parameter. In this sense, topological properties are often more stable with respect to the input parameters than geometrical properties, making topology potentially well-suited in a wide range of inference problems.

# 6 Discussion

We presented a novel Bayesian approach that allows for leveraging topological information in parameter estimation. Our method, which builds on the framework of generalised Bayesian inference, introduces a topology-based loss function into the construction of the posterior distribution. The resulting methodology is particularly suited for scenarios in which one has to deal with a 'black-box' likelihood function, i.e. where the likelihood is only accessible through simulations. This approach offers a theoretically grounded form of posterior belief that improves upon some of the challenges inherent in existing distance-based methods, which rely on geometrical features. The *main limitation* of our approach is that, although computational topology appears to be the appropriate tool for certain complex models, the resulting performance for parameter estimation highly depends on the setting (Lueckmann et al., 2021). A careful assessment of the given scenario is recommended beforehand. We restrict ourselves to scenarios with low-dimensional parameter spaces in this work, and leave the investigation of high-dimensional parameter spaces for future work. Moreover, we do not claim that our methodology outperforms benchmark Bayesian approaches under the assumption of a *known* likelihood function. However, we have empirically shown that our method outperforms geometrical approaches and summary statistics for parameter estimation in the setting of complex systems, which suggests future applications in other complex scenarios arising from life sciences data, for example.

Our work contributes to the ongoing evolution of Bayesian inference methods for complex models. It presents a new avenue of exploration that merges the advantages of both topological data analysis and Bayesian statistics. We believe that the principles and techniques introduced in this paper have broad applicability and offer a compelling new approach for uncertainty quantification in a wide range of data analysis tasks. From a machine learning perspective, the main novelty of this paper is to introduce a framework that fuses topology and Bayesian inference for scenarios in which the application of gradient-based approaches like neural networks constitutes a major obstacle. We anticipate future research to focus on further refining this method, specifically when it comes to calculating efficient estimates in sparse regimes. Moreover, we plan on assessing the performance arising from other topology-based formulations, which are geared towards specific modalities like meshes (Turner et al., 2014) or time series (Zeng et al., 2021).

### Reproducibility

We provide code for the given method and of the experiments shown in this paper under https://github.com/aidos-lab/TABAC.

### Acknowledgements

B.R. was partially supported by the Bavarian state government with funds from the *Hightech Agenda Bavaria*.

---

[5]For the parameter value of 0.3, we use the MSE loss as a performance measure, which is $\ll 0.001$ for each of the topological losses, and 0.211 for the geometrical loss.

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

Table 3: Estimating the radius parameters of a 2D sphere ($r$) and a 2D torus ($r, R$). For the more complex torus parameter estimation, we observe that our topological loss $\ell_T$ provides more reliable estimates than the geometrical loss $\ell_G$. The results show the mean of five runs for each parameter choice.

| | | Sphere radius $r$ | | | Torus radii $(r, R)$ | | | | | |
|---|---|---|---|---|---|---|---|---|---|---|
| *Loss* | *Sampling method* | 1.00 | 5.00 | 10.00 | $(1,2)$ | | $(3,5)$ | | $(5,10)$ | |
| $\ell_G$ | Importance sampling | 0.97 | 4.94 | 9.91 | 0.94 | 1.96 | 2.99 | 4.95 | 4.98 | 9.98 |
| | MCMC | 1.00 | 4.99 | 10.00 | 0.99 | 2.04 | 2.77 | 4.96 | 5.01 | 10.00 |
| $\ell_T$ | Importance sampling | 1.01 | 5.02 | 9.98 | 1.02 | 1.96 | 3.01 | 5.05 | 4.95 | 10.04 |
| | MCMC | 0.99 | 4.99 | 10.00 | 1.04 | 1.89 | 2.99 | 4.97 | 5.01 | 10.04 |

# A  Additional results

In the interest of reproducibility and to improve the understanding of our method, we also provide more detailed insights into our experiments.[6]

**Synthetic data parameter estimates.** As a consistency check and illustrative example, we use our proposed methods for inferring the radius parameters of two objects, namely the 2-sphere and the 2-torus. The 2-sphere is the surface of a closed 3-dimensional ball of radius $r$, while the 2-torus is constructed using an *inner radius r* and an *outer radius R*. The point clouds were sampled uniformly from the respective surface of known parameter.

Table 3 depicts the results of the inference process. We observe that both types of losses are capable of inferring the right parameters of these simple geometric objects, with our topology-based loss $\ell_T$ providing slightly more reliable estimates in the case of a torus. Table 3 shows the raw values of learning radius parameters for synthetic data sets (2D spheres and 2D tori). This experiment primarily shows that for such simple shapes, geometry-based and topology-based losses perform equivalently. For more complex data sets, however, we find that the improved robustness of our topology-based loss term helps in inferring the ground truth parameters.

**Swarm behaviour of the Vicsek Model.** Fig. 5 shows simulations of the Vicsek model for different noise parameters $\eta$. The higher $\eta$, the more 'chaotic' the behaviour of the resulting complex system.

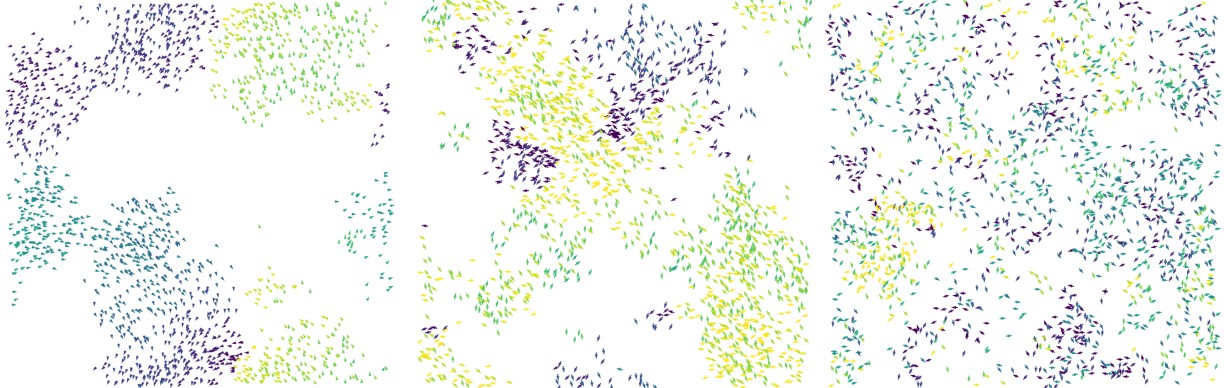

Figure 5: Left to right: simulations of the Vicsek model for $\eta = 0.15, 0.3$ and $0.6$, respectively. For lower values of $\eta$ swarming behaviour emerges, whereas for higher values of $\eta$ swarms tend to merge and the model becomes more ergodic.

---

[6]Certain parts of this section may have been developed with the use of OpenAI (2024).

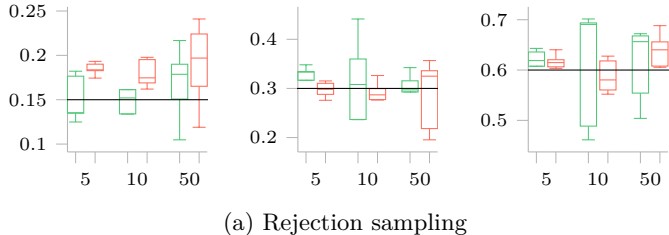

(a) Rejection sampling

Figure 6: Summary of parameter estimates for the noise parameter $\eta$ of the Vicsek model with respect to summary statistics and a standard rejection sampling procedure, using 5 repetitions of the inference procedure. Each subplot depicts estimates for $\eta \in \{0.15, 0.30, 0.60\}$, respectively. The true parameter is shown as a solid black line. The horizontal axis shows the number of iterations of the Vicsek model. Estimates based on the mean statistic $L_2$ loss and standard deviation statistic $L_2$ loss both are significantly less accurate than the respective topological loss, see Fig. 3b.

**Vicsek model parameter estimates.** Accompanying Fig. 3, we show the raw parameter estimates of $\eta$, the noise parameter of the model in Table 4. Moreover, we show the results of a *rejection sampling* procedure with summary statistics (mean and standard deviation, respectively), in Fig. 6. In the latter setting we used the $L_2$ distance between the respective summary statistics as the loss that determines the results of our method. We observe that there is no statistic that clearly discriminates the other one, and that the performance of the results highly depends on the parameter and time step that is considered. We find that the overall accuracy and reliability of the respective experiments by using topological losses is significantly higher; see Fig. 3 for a comparison.

Table 4: Noise parameter estimation after $n = \{5, 10, 50\}$ iterations of the Vicsek model. Estimates of $\eta$ based on a topological loss are always closer to the ground truth value than estimates obtained via a geometrical loss.

| | *Loss* | *Sampling method* | True parameter $\eta$ | | |
|---|---|---|---|---|---|
| | | | 0.15 | 0.30 | 0.60 |
| $n = 5$ | Hausdorff distance | Importance sampling | 0.17 | 0.35 | 0.67 |
| | | MCMC | 0.13 | 0.36 | 1.15 |
| | Topological distance | Importance sampling | 0.14 | 0.29 | 0.59 |
| | | MCMC | 0.12 | 0.33 | 0.63 |
| $n = 10$ | Hausdorff distance | Importance sampling | 0.11 | 0.39 | 0.70 |
| | | MCMC | 0.18 | 0.41 | 1.95 |
| | Topological distance | Importance sampling | 0.18 | 0.27 | 0.65 |
| | | MCMC | 0.13 | 0.34 | 0.62 |
| $n = 50$ | Hausdorff distance | Importance sampling | 0.12 | 0.29 | 0.72 |
| | | MCMC | 0.28 | 1.40 | 2.28 |
| | Topological distance | Importance sampling | 0.16 | 0.31 | 0.61 |
| | | MCMC estimate | 0.21 | 0.24 | 0.59 |

**Lattice Boltzmann model for fluid dynamics.** The degree of randomness in the local propagation and collision steps in the lattice Boltzmann method (LBM) model controls the turbulent behaviour of the global system, as is illustrated in Fig. 7. This is the parameter $\eta$ that we estimate in our experiments. The results are shown in Fig. 8 and Fig. 9, respectively. Using importance sampling, the topological loss outperforms

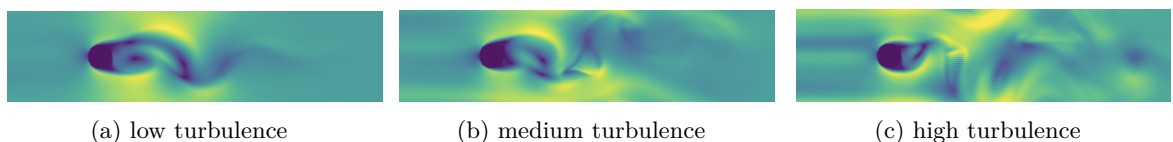

(a) low turbulence          (b) medium turbulence          (c) high turbulence

Figure 7: Left to right: simulations of the fluid model after 3000 iterations, for $\eta = 0.2, 0.3$ and $0.4$, respectively. For lower values of $\eta$ the fluid flow is less turbulent than for higher values of $\eta$.

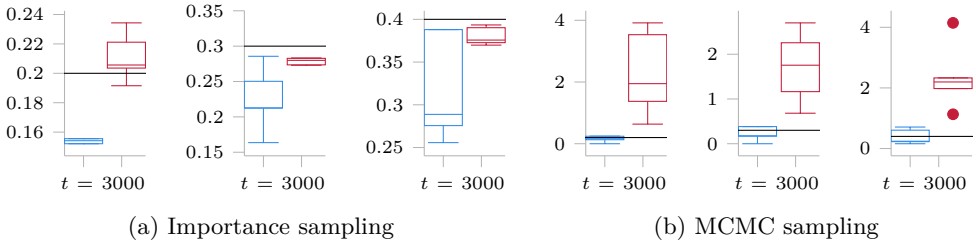

(a) Importance sampling                    (b) MCMC sampling

Figure 8: Summary of parameter estimates for the parameter $\eta$ of the fluid model, using 5 repetitions of the inference procedure. Each subplot depicts estimates for $\eta \in \{0.20, 0.30, 0.40\}$, respectively. The true parameter is shown as a solid black line. Estimates based on the geometrical loss $\ell_G$ are less accurate than the topological loss $\ell_T$.

the geometrical loss in terms of accuracy, as can be seen from the median and interquartile range of the boxplots in Fig. 8. Moreover, the topological loss in combination with importance sampling also outperforms the geometrical loss with MCMC sampling, as evidenced from Table 1. The topological loss in combination with MCMC sampling, however, is far off in terms of accuracy. The latter indicates that the convergence rates of MCMC sampling and importance sampling may be very different, and depend on the given setting. Finally, we note that rejection sampling with both the mean and standard deviation summary statistic does not perform reliably, and its accuracy highly depends on the true parameter in the underlying observation.

**Multi-scale structures in percolation models.** Fig. 10 shows an illustration of our proposed percolation model. The higher the value of $p$, the more likely it is for a pixel to be non-zero. Consequently, there are more non-zero pixels for higher values of $p$. As discussed in the main paper, the topological distance outperforms the SCC distance significantly, with both importance sampling and MCMC sampling. However, the rejection sampling procedure together with the mean statistic performs even more accurate in this experiment, as can be seen by comparing Fig. 11 and Fig. 12. This is not surprising: once the pixel size converges to infinity, the mean of the pixel values will converge to the expected pixel value, where the latter only depends on $p$ (since the maximum greyscale value is fixed in our experiments, and therefore does not have any discriminative

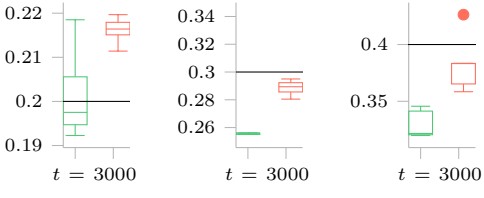

Rejection sampling

Figure 9: Summary of parameter estimates for the parameter $\eta$ of the fluid dynamics model with respect to summary statistics and a standard rejection sampling procedure, using 5 repetitions of the inference procedure. Each subplot depicts estimates for $\eta \in \{0.20, 0.30, 0.40\}$, respectively. The true parameter is shown as a solid black line. Estimates based on the mean statistic $L_2$ loss and on the standard deviation statistic $L_2$ loss tend to be less reliable than estimates based on the topological loss, see Fig. 8.

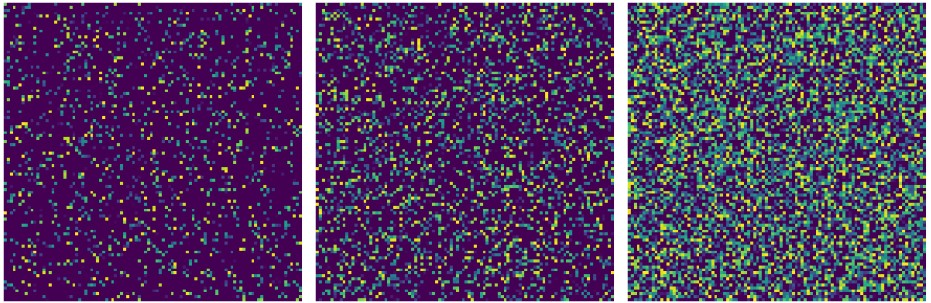

Figure 10: Left to right: samples of the proposed percolation model for $p = 0.15, 0.3$ and $0.6$, respectively. All samples admit a fixed maximum greyscale value of 50. Although the geometric distance between two samples of fixed $p$ can be large due to the uniform sampling of pixels, the overall topological structure (which is captured by persistent homology) for a given $p$ is more stable with respect to sampling.

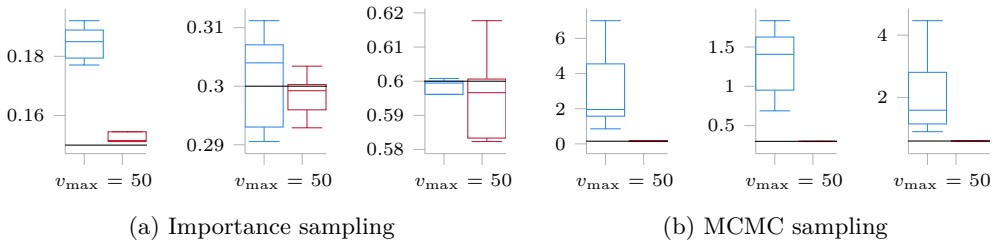

(a) Importance sampling          (b) MCMC sampling

Figure 11: Summary of parameter estimates for the parameter $p$ of the Percolation model, using 5 repetitions of the inference procedure. Each subplot depicts estimates for $p \in \{0.15, 0.30, 0.60\}$, respectively. The true parameter is shown as a solid black line. Estimates based on the geometrical loss $\ell_G$ are less accurate than the topological loss $\ell_T$, in particular for larger number of iterations.

power for inference). Therefore, at least in a large-pixel regime we can infer the true parameter of the observation from the mean of its pixel values. Note that this is due to the additional structure in this experiment, and does not hold for complex systems that cannot be described in such a simple way, as has been seen in the previous experiments. Finally, rejection sampling with the standard deviation statistic leads to highly erroneous results, as can also be seen from Fig. 12. For complex systems it is therefore very difficult to determine the 'right' summary statistic which contains a sufficient amount of information to infer the underlying parameter. This choice is not necessary when using topological losses, which makes the latter an appropriate generic choice, in many applications.

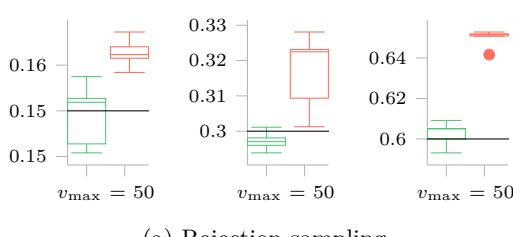

(a) Rejection sampling

Figure 12: Summary of parameter estimates for the parameter $p$ of the Percolation model with respect to summary statistics and a standard rejection sampling procedure, using 5 repetitions of the inference procedure. Each subplot depicts estimates for $p \in \{0.15, 0.30, 0.60\}$, respectively. The true parameter is shown as a solid black line. Estimates based on the mean statistic $L_2$ loss are more accurate than the standard deviation statistic $L_2$ loss.

Table 5: For 500 simulation points: Estimating the randomness parameter of the lattice Boltzmann model after 3000 simulation iterations, using 5 repetitions of the inference procedure. The topological loss together with importance sampling outperforms all of the other configurations.

| | | Randomness parameter $\eta$ | | |
|---|---|---|---|---|
| *Loss* | *Sampling method* | 0.20 | 0.30 | 0.40 |
| $\ell_G$ | Importance sampling | $0.14 \pm 0.02$ | $0.22 \pm 0.02$ | $0.26 \pm 0.03$ |
| | MCMC | $0.18 \pm 0.07$ | $0.21 \pm 0.03$ | $0.22 \pm 0.07$ |
| $\ell_T$ | **Importance sampling** | $\mathbf{0.21 \pm 0.01}$ | $\mathbf{0.28 \pm 0.01}$ | $\mathbf{0.38 \pm 0.05}$ |
| | MCMC | $2.78 \pm 1.82$ | $2.91 \pm 2.11$ | $2.35 \pm 0.80$ |

**Ablation study**

# B Further details and proof

---
**Algorithm 2** MCMC approximation of the target
---

1: **Input:** Observed data $y$, start value $\theta_0$, proposal distribution $q(\vartheta \,|\, \theta)$
2: Sample $X_0 \sim p(x \,|\, \theta_0)$
3: **for** $i = 1 : K$ **do**
4:     Sample $U \sim \mathsf{Unif}[0, 1]$
5:     Sample $\theta' \sim q(\cdot \,|\, \theta_{i-1})$
6:     Sample $X' \sim p(x \,|\, \theta')$
7:     Compute

$$a(\theta_{i-1}, \theta') = \frac{e^{-\ell(y, X')} p(\theta')}{e^{-\ell(y, X_{i-1})} p(\theta_{i-1})} \frac{q(\theta_{i-1} \,|\, \theta')}{q(\theta' \,|\, \theta_{i-1})}$$

8:     **if** $U \leq a(\theta_{i-1}, \theta')$ **then**
9:         Set $\theta_i = \theta', X_i = X'$
10:     **else**
11:         Set $\theta_i = \theta_{i-1}, X_i = X_{i-1}$
12:     **end if**
13: **end for**
14: **Output:**

$$\boldsymbol{\theta} = (\theta_{1:K}).$$

---

**Description of Algorithm 2**

**Inputs:**

- **Observed Data** $y$**:** The actual data you have collected.

- **Start Value** $\theta_0$**:** An initial guess for the parameter $\theta$, whose posterior distribution one intends to estimate.

- **Proposal Distribution** $q(\vartheta \,|\, \theta)$**:** A distribution used to propose new values for the parameter $\theta$, based on its current value.

**Steps:**

**Initialize:** Start by sampling $X_0$ from the distribution $p(x \,|\, \theta_0)$, where $p(x \,|\, \theta)$ represents the likelihood of the data $x$ given a parameter $\theta$. This initializes the first sample of the latent variable $X_0$.

**Iterative Sampling (Loop from $i = 1$ to $K$):** For each iteration, repeat the following steps to generate a sequence of samples that approximate the distribution of $\theta$:

- **Generate Random Number U:** Sample $U \sim \mathsf{Unif}[0, 1]$. This will be used for the acceptance/rejection step.

- **Propose a New Parameter Value $\theta'$:** Sample a new candidate value $\theta'$ from the proposal distribution $q(\cdot \mid \theta_{i-1})$, which suggests a new possible value for the parameter based on the current value $\theta_{i-1}$.

- **Sample a New Latent Variable $X'$:** Sample a new value for the latent variable $X'$ from the distribution $p(x \mid \theta')$, corresponding to the newly proposed parameter $\theta'$.

- **Compute the Acceptance Ratio $a(\theta_{i-1}, \theta')$:** Calculate the acceptance ratio, which compares the likelihood of the data given the new parameter $\theta'$ and the current parameter $\theta_{i-1}$, adjusted by their prior probabilities and proposal distributions. This ratio helps decide whether the new proposed value $\theta'$ should be accepted or not.

- **Acceptance/Rejection Step:** Compare $U$ (the random number sampled earlier) with the acceptance ratio $a(\theta_{i-1}, \theta')$:
  - If $U \leq a(\theta_{i-1}, \theta')$, accept the new proposed parameter $\theta'$ and set $\theta_i = \theta'$, and also update the latent variable to $X_i = X'$.
  - Otherwise, reject the proposed value and retain the current parameter and latent variable, i.e., set $\theta_i = \theta_{i-1}$ and $X_i = X_{i-1}$.

**Repeat:** Continue this process for a total of $K$ iterations.

**Output:** After completing the iterations, the algorithm outputs a sequence of sampled parameter values $\boldsymbol{\theta} = (\theta_1, \theta_2, \ldots, \theta_K)$, which approximates the posterior distribution of the parameter given the observed data.

**Proof about the comparison-based posterior**

**Proposition B.1.** *The comparison-based posterior*

$$\pi(\theta, x \mid y) = \frac{\exp\left(-\ell(y, x)\right) p(x \mid \theta) p(\theta)}{\int_{\mathsf{X}} \int_{\Theta} \exp\left(-\ell(y, x)\right) p(x \mid \theta) p(\theta) \mathrm{d}\theta \mathrm{d}x}$$

*is the solution $q^*(\theta, x) = \pi(\theta, x \mid y)$ to the optimisation problem*

$$q^* = \arg\min_{q \in \mathcal{P}(\Theta \times \mathsf{X})} \left\{ \mathbb{E}_q[\ell(y, x)] + \mathrm{KL}\left(q(\theta, x), p(x \mid \theta) \pi(\theta)\right) \right\}.$$

*Proof.*

$$\begin{aligned}
q_B &= \arg\min_q \left\{ \mathbb{E}_q\left[\ell(y, x)\right] + KL\left(q(x, \theta), p(x \mid \theta) \pi(\theta)\right) \right\} \\
&= \arg\min_q \left\{ \iint \ell(y, x) q(x, \theta) \mathrm{d}x \mathrm{d}\theta + \iint \log \frac{q(x, \theta)}{p(x \mid \theta) \pi(\theta)} q(x, \theta) \mathrm{d}x \mathrm{d}\theta \right\} \\
&= \arg\min_q \left\{ \iint \log\left(\exp \ell(y, x)\right) q(x, \theta) \mathrm{d}x \mathrm{d}\theta + \iint \log \frac{q(x, \theta)}{p(x \mid \theta) \pi(\theta)} q(x, \theta) \mathrm{d}x \mathrm{d}\theta \right\} \\
&= \arg\min_q \left\{ \iint \log\left(\frac{q(x, \theta)}{p(x \mid \theta) \pi(\theta) \exp\left(-\ell(y, x)\right)}\right) q(x, \theta) \mathrm{d}x \mathrm{d}\theta \right\} \\
&= \arg\min_q \left\{ \iint \log\left(\frac{q(x, \theta)}{p(x \mid \theta) \pi(\theta) \exp\left(-\ell(y, x)\right) Z^{-1}}\right) q(x, \theta) \mathrm{d}x \mathrm{d}\theta \right\} - \log Z \\
&= \arg\min_q KL\left\{ q(x, \theta), p(x \mid \theta) \pi(\theta) \exp\left(-\ell(y, x)\right) Z^{-1} \right\}.
\end{aligned}$$

Hence, the minimiser $q$ is such that

$$q(x, \theta) = \frac{p(x \mid \theta)\pi(\theta)\exp\left(-\ell(y, x)\right)}{Z}, \quad q(\theta) = \int q(x, \theta)\mathrm{d}x$$

and

$$Z := \iint \exp\left(-\ell(y, x)\right) p(x \mid \theta)\pi(\theta)\mathrm{d}x\mathrm{d}\theta.$$

$\square$

**Homology in a nutshell** Homology groups are a fundamental tool in algebraic topology, providing algebraic representations of the topological structure of a space. Given a topological space $K$, the homology group $H_d(K)$ captures the $d$-dimensional topological features of the space, such as connected components, loops, and voids. At a high level, the homology group $H_d(K)$ encodes the $d$-dimensional 'holes' of the space. $H_0(K)$ represents the number of connected components (0-dimensional holes). $H_1(K)$ captures loops or 1-dimensional holes, such as circular paths that do not bound any region. Higher-dimensional homology groups $H_d(K)$ with $d \geq 2$ detect higher-dimensional voids, such as cavities inside the space. Homology groups are defined through an algebraic process. The space $K$ is decomposed into simpler components, such as simplices in the case of simplicial complexes. Chains are formed as formal combinations of these components, and boundary operators are introduced to map a $d$-dimensional simplex to its $(d-1)$-dimensional boundary. The homology group $H_d(K)$ is then given by the quotient:

$$H_d(K) = \ker(\partial_d)/\mathrm{im}(\partial_{d+1}),$$

where $\partial_d$ denotes the boundary operator at dimension $d$. Intuitively, elements of $H_d(K)$ are equivalence classes of $d$-dimensional cycles (chains with no boundary) in $K$ that are not boundaries of higher-dimensional objects. In summary, the homology group $H_d(K_i)$ encodes critical information about the $d$-dimensional features of a space $K$, allowing to analyze and compare spaces using algebraic methods.

**Pseudo-marginal MCMC in a nutshell** Pseudo-marginal Markov Chain Monte Carlo (MCMC) methods are a class of algorithms used for sampling from a posterior distribution when the likelihood function is intractable or expensive to compute. In such scenarios, the exact likelihood is replaced by an unbiased estimator, which introduces randomness into the acceptance ratio of the Metropolis-Hastings (MH) algorithm. Despite this randomness, pseudo-marginal MCMC methods can still produce samples from the correct target distribution. In Bayesian inference, one often needs to sample from a posterior distribution $\pi(\theta \mid y) \propto \pi(\theta)p(y \mid \theta)$, where $\pi(\theta)$ is the prior, and $p(y \mid \theta)$ is the likelihood function. In many practical cases, such as when dealing with complex models, the likelihood $p(y \mid \theta)$ may not be analytically tractable or computationally feasible to evaluate directly. Pseudo-marginal MCMC provides a solution by replacing the likelihood with an unbiased estimator $\hat{p}(y \mid \theta)$. This approach retains the correct target distribution $\pi(\theta \mid y)$ because the pseudo-marginal method treats the noisy estimate $\hat{p}(y \mid \theta)$ as if it were the exact likelihood within the Metropolis-Hastings framework. The pseudo-marginal MCMC algorithm proceeds similarly to the standard Metropolis-Hastings algorithm, with the key difference being that the likelihood ratio in the acceptance probability is replaced by a ratio of likelihood estimators. Specifically, given a current state $\theta_t$, a new state $\theta'$ is proposed, and the acceptance probability is calculated as:

$$\alpha(\theta_t, \theta') = \min\left(1, \frac{\hat{p}(y \mid \theta')\pi(\theta')q(\theta_t \mid \theta')}{\hat{p}(y \mid \theta_t)\pi(\theta_t)q(\theta' \mid \theta_t)}\right),$$

where $\hat{p}(y \mid \theta')$ and $\hat{p}(y \mid \theta_t)$ are unbiased estimates of the likelihood, and $q(\theta' \mid \theta_t)$ is the proposal distribution. Crucially, the unbiasedness of the likelihood estimator ensures that, despite the randomness introduced, the chain remains reversible and converges to the correct posterior distribution $\pi(\theta \mid y)$. See Andrieu & Roberts (2009) for a thorough introduction to pseudo-marginal MCMC.

