# OpenReview forum: "Bayesian Computation Meets Topology"
_TMLR — Accepted by TMLR_

### Review · Reviewer_m291 · 2024-08-12

**Summary Of Contributions:**

In this paper, the authors present a methodology to perform Bayesian inference (i.e., sampling from a posterior distribution) with data topological knowledge, which is a setting that is still largely unexplored in model parameter estimation. This approach is specifically designed for scenarios where the likelihood of the statistical model is only available via simulations (simulation-based inference). In practice, these simulations are given by a generative model; here, the likelihood of this model is also assumed to be non-differentiable with respect to the parameter, which makes impossible to use gradient-based approaches such as variational inference.

The proposed method builds on the framework of generalized Bayesian inference by introducing a topological-based loss function in the construction of a two-state posterior distribution (combining model parameter and simulated data). This loss is computed as the Wasserstein-2 distance between sets of multi-scale topological features obtained from real data and simulated data (namely, Victoris-Rips complexes and cubical complexes, depending on the application). This consists in the main contribution of the paper. Once the generalized posterior is built, Bayesian inference is conducted by directly sampling from this distribution either via importance sampling or Metropolis-Hastings MCMC method. This approach is supported by numerical experiments conducted on complex physics systems, which compare the use of topological-based loss vs geometry-based loss under the same budget of simulations. In all of the presented settings, the topological-based approach is on par or outperforms the geometry-based approach.

**Audience:**

Yes

**Broader Impact Concerns:**

None.

**Claims And Evidence:**

Yes

**Requested Changes:**

**Critical**:
- In my opinion, **an ablation study on the number of simulations** in each numerical setting is missing. This study should inform how robust the proposed method is with respect to the sampling complexity, and how well it performs when increasing this budget. For instance, plotting the parameter estimation error vs the sampling complexity could be a good idea.
- The **limitation to low dimensional cases** (here, I am referring to the parameter dimension) should be addressed either by performing new numerical experiments in higher dimension or clearly stating this limitation in the paper.
- The **hyperparameters of the method should be clearly stated** in each numerical experiment (see the 'Weaknesses' section), and tested with small ablation studies to validate robustness. For instance, with good tuning of the Markov kernel in the MCMC algorithm, I am quite convinced that the MCMC sampling could be as good as importance sampling in each setting.
- More details on **the computation of the Wasserstein-2 distance** should be given: complexity in practice (which is not trivial at all), preference over regularized Wasserstein-2 distance.

**Moderate**:
- The field name 'Simulation-based inference' with related work should appear in the introduction, as it encompasses the scenario chosen by the authors for Bayesian inference.
- The Algorithm 2 should be explained by sentences which give the main infos, i.e., computing a MH rate on a two-state distribution, making explicit the two-state proposal Markov kernel (give advantages and drawbacks), explaining why the MH rate simplifies in $x$... As such, it is not commentated at all, which may be hard to understand for non-expert readers.

**Strengths And Weaknesses:**

**Strengths**:
- The paper is overall well written, easy to understand.
- The use of topological-based loss function in the generalized Bayesian inference is of high interest for complex physics systems, to build an accurate posterior distribution.
- The explanation of the topological tools is accessible for non-expert readers.
- The choice of the topological complexes in each numerical setting is well justified.

**Weaknesses**:
- The numerical experiments are limited to very-low dimensional settings (only 1D for complex systems, 2D for synthetic data). It is unclear to state if the proposed method may scale to higher dimensional settings under reasonable amount of simulations in the sampling phase (in particular, I am quite concerned about the IS approach which is known to scale poorly with dimension). Note that this is not indicated as a limitation of the method.
- The overall computational budget of the method is unclear to me: in particular, the topological based approach requires to compute a Wasserstein-2 distance which is known to scale very poorly with respect to the data dimension and to the number of points. To ensure fair comparison with the geometry-based approach, the cost of the loss computation should also appear.
- The dependence to hyperparameters is unclear: especially, about the number of scales in the topological part and the design of the Markov kernel on $\theta$ in the MCMC setting (which is highly crucial).
- It is not clear why the exact Wasserstein distance is preferred to the entropy-regularized Wasserstein distance (which is known to be more efficient in practice).

---

> ### Author Response · Authors · 2024-08-21
> **Thank you for your feedback/ First clarifications**
>
> Dear Reviewer,
>
> thank you for carefully reviewing our paper, and for acknowledging that it is of high interest!
>
> > The overall computational budget of the method is unclear to me: in particular, the topological based approach requires to compute a Wasserstein-2 distance which is known to scale very poorly with respect to the data dimension and to the number of points. To ensure fair comparison with the geometry-based approach, the cost of the loss computation should also appear.
>
> Notice that the Wasserstein distance is evaluated on the persistence information, which is always 2 (independently of the data dimension). However, the computational complexity of the persistence information itself depends on the intrinsic dimension of the data (and the number of points). We are happy to add a discussion on the computational complexity of the method, in a revision, for both topological and geometric losses.
>
> > The dependence to hyperparameters is unclear: especially, about the number of scales in the topological part and the design of the Markov kernel on θ in the MCMC setting (which is highly crucial).
>
> Regarding the number of scales in the topological part there is no choice involved since we use the whole persistence diagram and therefore consider topological features at all scales. Regarding the Markov kernel the only hyperparameters involved are the ones concerning the prior distribution. We hope that this addresses your questions, and we are happy to give details on these points in our revised paper!
>
> > It is not clear why the exact Wasserstein distance is preferred to the entropy-regularized Wasserstein distance (which is known to be more efficient in practice).
>
> Indeed, under the hood our implementation uses a regularized version of the Wasserstein distance, as suggested by you. We will explain this aspect in our revision!
>
> > In my opinion, an ablation study on the number of simulations in each numerical setting is missing. This study should inform how robust the proposed method is with respect to the sampling complexity, and how well it performs when increasing this budget. For instance, plotting the parameter estimation error vs the sampling complexity could be a good idea.
>
> In all our experiments we used 250 simulations in each run since we found that increasing the number of simulations did not lead to improved performance, for both importance sampling and MCMC sampling. However, we are happy to provide more details on this aspect in a revision.
>
> > The limitation to low dimensional cases (here, I am referring to the parameter dimension) should be addressed either by performing new numerical experiments in higher dimension or clearly stating this limitation in the paper.
>
> We do not have concrete domain expertise or strong benchmarks that require higher-dimensional parameter spaces, hence, as per the reviewer’s suggestion, we will indeed note this limitation and restrict ourselves to the low-dimensional regime for now. Would this be acceptable?
>
> > The field name 'Simulation-based inference' with related work should appear in the introduction, as it encompasses the scenario chosen by the authors for Bayesian inference.
>
> > The Algorithm 2 should be explained by sentences which give the main infos, i.e., computing a MH rate on a two-state distribution, making explicit the two-state proposal Markov kernel (give advantages and drawbacks), explaining why the MH rate simplifies in
> ... As such, it is not commentated at all, which may be hard to understand for non-expert readers.
>
> We will address these two points in our revision.
>
> Thank you once again for your feedback! In the meantime, please let us know if there are any other questions or concerns. We are happy to clarify them!
>
> Best regards,
>
> the Authors

---

### Review · Reviewer_WZUE · 2024-08-18

**Summary Of Contributions:**

This paper proposes a novel approach for performing Bayesian inference using topology-based loss functions. The authors develop a framework to leverage topological information in parameter estimation through a comparison-based posterior approach. They introduce topology-based loss functions that can be used within this Bayesian framework and demonstrate how sampling algorithms can utilize these topologically-informed loss functions for Bayesian inference.

**Audience:**

Yes

**Broader Impact Concerns:**

The paper does not raise any significant ethical concerns.

**Claims And Evidence:**

Yes

**Requested Changes:**

1. Provide a detailed explanation of what distinguishes this work from existing approaches. Highlight the novelty of the methodology beyond the application of topological losses in the comparison-based posterior framework.

2. Include comparisons to relevant baseline methods to demonstrate the added value of the proposed approach. Additionally, perform a more comprehensive analysis of the uncertainty quantification capabilities, including comparisons with traditional methods.

3. Expand on the reasoning behind why standard gradient-based methods cannot be applied to the specific models discussed. This should include a more thorough discussion of the limitations of traditional approaches in these settings.

4. Include a discussion of alternative methods for parameter estimation in complex models, positioning the proposed approach within the broader context of existing research. This will help to better frame the contribution of the paper.

5. Improve the clarity of technical explanations, particularly in sections that may be challenging for readers unfamiliar with topology or Bayesian inference, such as the discussion on pseudo-marginal MCMC.

**Strengths And Weaknesses:**

Strengths:

1. The paper presents a theoretically grounded method for integrating topological information into Bayesian inference, particularly useful for complex models where traditional likelihood-based approaches are intractable.

2. It demonstrates enhanced performance over geometric losses in multiple simulation studies, highlighting the potential of topology-based losses for accurate parameter estimation.

Weaknesses:

1. The novelty of the paper is a concern, as it primarily applies existing comparison-based posterior approaches with topological losses, rather than introducing fundamentally new concepts.

2. The paper lacks baselines and does not include comparisons to alternative methods for parameter estimation within these settings.

3. The motivation and justification for not using standard gradient-based approaches are unclear, leaving the reader questioning the necessity of the proposed method.

---

> ### Author Response · Authors · 2024-08-21
> **Thank you for your feedback/ First Clarifications**
>
> Dear Reviewer,
>
>
> thank you for carefully reading and reviewing our paper!
>
> > The novelty of the paper is a concern, as it primarily applies existing comparison-based posterior approaches with topological losses, rather than introducing fundamentally new concepts.
>
> In our opinion, the main novelty consists of incorporation of topological information into parameter inference and its utility for parameter estimation in complex systems. Notice that using information about the overall structure of the data (such as its shape) for parameter inference is not straight-forward, although it may be highly beneficial (as shown in this paper). We are happy to further clarify this aspect in a revision!
>
> > The paper lacks baselines and does not include comparisons to alternative methods for parameter estimation within these settings.
>
> The fact that our experiments take place in a possibly non-differentiable setting (due to both the topological loss and the simulation/data model itself) impedes a comparison against standard baseline models such as neural network approaches. However, we are happy to add a comparison against a baseline rejection sampling procedure.
>
> > The motivation and justification for not using standard gradient-based approaches are unclear, leaving the reader questioning the necessity of the proposed method.
>
> As explained in the introductory section, standard gradient-based methods may not be applicable in our setting, for example since the likelihood is non-differentiable. For instance, swarming models are known to admit tipping points that lead to non-continuous (and therefore non-differentiable) behavior in the parameters. Also standard methods do not allow for an assessment of uncertainty which is naturally implemented in our method. In our revised version, we will highlight this aspect better.
>
> > Provide a detailed explanation of what distinguishes this work from existing approaches. Highlight the novelty of the methodology beyond the application of topological losses in the comparison-based posterior framework.
>
> We are happy to provide a more detailed explanation of what distinguishes this work from existing approaches, in a revision.
>
> > Include a discussion of alternative methods for parameter estimation in complex models, positioning the proposed approach within the broader context of existing research. This will help to better frame the contribution of the paper.
>
> We are unaware of other existing methods that permit the inclusion of a topology-based loss. We will highlight this in our revision. Moreover, we would be grateful if the reviewer could point us towards alternative methods that they consider suitable in this context. In the meantime, we will also include standard rejection sampling techniques, for instance, thus highlighting the utility of our proposed method. Would this be deemed acceptable?
>
> > Improve the clarity of technical explanations, particularly in sections that may be challenging for readers unfamiliar with topology or Bayesian inference, such as the discussion on pseudo-marginal MCMC.
>
> We will give more background on the methods used in this paper in our revision.
>
> Thank you once again for your feedback! In the meantime, please let us know if there are any other questions or concerns. We are happy to clarify them!
>
> Best regards,
>
> the Authors

---

### Review · Reviewer_KcrH · 2024-08-24

**Summary Of Contributions:**

This work lies at the intersection of Bayesian inference and topology. The authors propose a new generalized Bayesian inference approach that uses topological loss functions in parameter estimation. The topological loss can accommodate different data modalities and result in more accurate parameter inference. The authors demonstrate the effectiveness of their approach on three example problems, including the Vicsek model, Lattice Boltzmann Simulation and Percolation Model, and showed their method is more accurate than the geometrical loss function.

**Audience:**

Yes

**Claims And Evidence:**

Yes

**Requested Changes:**

- In Prop. 3.1., $\pi$ seems abused. Both $\pi(\theta, x \mid y)$ and $\pi(\theta)$ are used. The first represents the optimal solution. The second is used in the objective function. What does $\pi$ denote exactly?

- I am a bit confused by the challenges the authors want to address with their method, which combines topology and Bayesian inference. Initially, my understanding of the targeting problem is cases when the likelihood function is not analytically tractable or differentiable. However, it seems this has already been addressed by generalized Bayesian methods. So what is the exact benefit of introducing topological loss functions? And what is the main challenge that the authors are trying to address?

- Furthermore, the application scenario of the proposed method is not clear. In Introduction and Method sections, the authors mentioned analytically intractable or non-differentiable likelihoods. In conclusion part, the authors mentioned unknown or "black-box" likelihood. Additionally, the authors also mentioned "complex systems". In contrary to all these, the problems in experiments seem to have known likelihood functions that can be expressed explicitly, thus tractable. So I am a little confused. What is the exact application scenario of the proposed method?

- In Vicsek model, why is the likelihood not analytically tractable? Based on my understanding, the distribution of the position of each agent can be written in closed form at any time step.

- Can the authors clarify in the experiments section whether the likelihood is intractable or non-differentiable for Lattice Boltzmann Simulation and Percolation Model? Why is it necessary or beneficial to use topological loss in these cases?

- I am a bit confused by Section 3.2.
  - "In the ensuing discussion, we will illustrate that a modification of the aforementioned approach, specifically," Does the aforementioned approach refer to the generalized Bayesian method in Eq. (1)?
  - "This approach, however, encounters several obstacles." What does "this approach" refer to? Is it a naive, intermediate approach proposed by the authors that inspired the final approach? Or is it an existing approach in the literature?
  - The motivation of Prop. 3.1 is a bit unclear. Is Prop. 3.1 a new form of generalized Bayesian posterior? How is it related to the overarching goal of incorporating topolocial information into Bayesian inference?

- In the experiments, the authors only compared with the geometrical loss function. Is any of the methods mentioned in related works applicable to compare?

- In Table 2, how did the authors define "outperform"? When p=0.3, geometrical loss with importance sampling achieved higher accuracy but much larger variance than topological loss ($0.30 \pm 0.47$ vs. $0.29 \pm 0.00$). The authors may want to clarify their consideration here, how they determine which is better.

**Strengths And Weaknesses:**

**Strengths**
- The paper is very well-written, with barely any grammatical errors.
- Good references and theoretical comparisons to related work.

**Weaknesses**
- The definition of the problem and contribution can be more clear, especially the specific challenge addressed and the benefits brought by tological loss functions.
- The application scenario can be more clear. The paper mentions various scenarios (intractable/non-differentiable likelihoods, unknown/"black-box" likelihoods, complex systems, etc.). However, a precise definition seems missing.
- Comparison method is relatively limited.

---

> ### Author Response · Authors · 2024-08-26
> **Thank you for your feedback/ First clarifications**
>
> Dear reviewer,
>
> thank you for carefully reading our work!
>
> > The definition of the problem and contribution can be more clear, especially the specific challenge addressed and the benefits brought by tological loss functions.
>
> Thank you for this suggestion. We will add more details on the problem statement and the benefits brought by topological losses in our revision!
>
> > The application scenario can be more clear. The paper mentions various scenarios (intractable/non-differentiable likelihoods, unknown/"black-box" likelihoods, complex systems, etc.). However, a precise definition seems missing.
>
> We apologize for the confusion. We particularly focus on scenarios where the likelihood function is analytically intractable, and therefore exact Bayesian inference is impossible. However, we assume that we have access to a data simulating model which allows for simulation-based Bayesian inference. One important class of such models are agent-based models (e.g. the Vicsek model) which are generally known to admit non-differentiable likelihood functions (e.g. due to tipping points in the case of the Vicsek model).  We will clarify this in a revision.
>
> > Comparison method is relatively limited.
> In our view, the main focus of the paper is to compare geometrical losses against topological losses in order to justify that using topological losses can be beneficial for inference, in many cases. However, we are happy to add a comparison against standard rejection sampling procedures, would this be acceptable?
>
> > In Prop. 3.1., π seems abused. Both π(θ,x∣y) and π(θ) are used. The first represents the optimal solution. The second is used in the objective function. What does π denote exactly?
>
> Thank you for pointing this out! In fact, the second one (in the objective function) is a typo and should denote the prior, p(θ). We will fix this in our revision!
>
> > I am a bit confused by the challenges the authors want to address with their method, which combines topology and Bayesian inference. Initially, my understanding of the targeting problem is cases when the likelihood function is not analytically tractable or differentiable. However, it seems this has already been addressed by generalized Bayesian methods. So what is the exact benefit of introducing topological loss functions? And what is the main challenge that the authors are trying to address?
>
> Thank you for this question! Analytically intractable or non-differentiable likelihood functions are rather a description of the scenario we focus on. The reason of introducing a topological loss functions lies in the fact that topological descriptors (i.e. “the shape”) of the underlying data provides information which is highly beneficial for inference (we believe that this can be seen from the experiments, but we will improve our discussion here). Our interpretation is that altering the parameters to be inferred leads to a change in the overall structure (i.e. shape) of the data which cannot readily be captured via geometrical losses (see e.g. Fig.3, which shows the results for the Vicsek model). Finally, simulation-based inference provides a natural way of incorporating this crucial structural information, by using (topological) losses. We hope this answers your question, we are happy to add these aspects in our revision!
>
> > Furthermore, the application scenario of the proposed method is not clear. In Introduction and Method sections, the authors mentioned analytically intractable or non-differentiable likelihoods. In conclusion part, the authors mentioned unknown or "black-box" likelihood. Additionally, the authors also mentioned "complex systems". In contrary to all these, the problems in experiments seem to have known likelihood functions that can be expressed explicitly, thus tractable. So I am a little confused. What is the exact application scenario of the proposed method?
>
> We focus on scenarios where the likelihood function is analytically unknown, but where simulating data from the likelihood is in fact possible. Although this might seem counterintuitive at first glance, there are a plethora of models which operate in this setting. One famous example is the Vicsek model where the behavior of the single agents leads to highly complex collective behavior that cannot be expressed analytically, in order to perform inference. However, it is indeed possible to run this simulation model (and therefore to sample data from the likelihood).
>
> > In Vicsek model, why is the likelihood not analytically tractable? Based on my understanding, the distribution of the position of each agent can be written in closed form at any time step.
>
> Although the position of each agent can be written down analytically, the dependencies between agents leads to highly complex behavior of the overall system which cannot be expressed analytically. We will clarify this in our revision. (1/n)

---

> > ### Author Response · Authors · 2024-08-26
> > **Continuation of response**
> >
> > > Can the authors clarify in the experiments section whether the likelihood is intractable or non-differentiable for Lattice Boltzmann Simulation and Percolation Model? Why is it necessary or beneficial to use topological loss in these cases?
> >
> > For both the Lattice Boltzmann and percolation model (and the Vicsek model), we found that the topological information incorporated via the topological loss leads to better inference accuracy. We will further clarify this in our revision!
> >
> > > "In the ensuing discussion, we will illustrate that a modification of the aforementioned approach, specifically," Does the aforementioned approach refer to the generalized Bayesian method in Eq. (1)?
> >
> > Here,  “aforementioned approach” refers to “Uncertainty estimates predicated on distance functions have a long-standing history in Bayesian statistics as a means of approximating posterior distributions, with notable use in the realm of approximate Bayesian computation (ABC) (Tavaré et al., 1997; Pritchard et al., 1999; Beaumont et al., 2002).”.
> >
> > > "This approach, however, encounters several obstacles." What does "this approach" refer to? Is it a naive, intermediate approach proposed by the authors that inspired the final approach? Or is it an existing approach in the literature?
> >
> > Here, “this approach” refers to  “a generalised posterior from ABC such as Schmon et al. (2021)” (which leads to Eq.4). In this sense, the cited work inspired our final approach.
> >
> > > The motivation of Prop. 3.1 is a bit unclear. Is Prop. 3.1 a new form of generalized Bayesian posterior? How is it related to the overarching goal of incorporating topolocial information into Bayesian inference?
> >
> > Prop 3.1 shows that the comparison-based posterior of Eq.6 is in fact a solution to the optimization problem at hand (which is the analogue of Eq.1 in our setting). The motivation is therefore to use comparison-based posteriors for approximate inference under the assumption that an appropriate loss function is available. Prop 3.1 does not necessitate a topological loss and can be seen independently of a specific loss function. However, in particular it allows for incorporating topological information via topological losses which is not straight-forward by using traditional approaches.
> >
> > The argument and proof that comparison-based posteriors can be seen as the solution to a generalized Bayesian optimization problem is indeed new. While distance-based loss functions, where a parameter $\theta$ is only available through simulation have been used before, they are commonly seen as approximation of an idealized loss function
> > $$
> > \ell(y, \theta) := \lim_{n\rightarrow \infty} \sum_{i=1}^n \ell (y, x_i).
> > $$
> >
> > The motivation of Prop. 3.1 is to demonstrate that comparison-based posteriors do not have to be seen as a “suboptimal” approximation to an abstract ideal, but can be seen as a valid generalized posterior in their own right.
> >
> > > In the experiments, the authors only compared with the geometrical loss function. Is any of the methods mentioned in related works applicable to compare?
> >
> > In our view, the focus of this work is to show that topology incorporated via loss functions can be a crucial ingredient for inference, which cannot be done by using geometrical losses. Therefore, a comparison between topological and geometrical losses was at the core of this paper. Moreover, many of the inference procedures are not applicable due to likelihood constraints as mentioned in the Introduction. However, we are happy to add a comparison against a rejection sampling procedure, in our revision!
> >
> > > In Table 2, how did the authors define "outperform"? When p=0.3, geometrical loss with importance sampling achieved higher accuracy but much larger variance than topological loss (0.30±0.47 vs. 0.29±0.00). The authors may want to clarify their consideration here, how they determine which is better.
> >
> > We would suggest utilizing an MSE loss for such an assessment. We are happy to give the details in our revision!
> >
> > Thank you once more for your engagement and your feedback. In the meantime, please let us know if there are any other concerns or questions. We are happy to clarify them!
> >
> > Best wishes,
> > the Authors

---

> > > ### Comment · Reviewer_KcrH · 2024-08-27
> > > **Reviewer Comment**
> > >
> > > Thank the authors for their swift responses. They have cleared most of my confusions. I am looking forward to the revised paper.

---

> > > > ### Comment · Reviewer_KcrH · 2024-09-20
> > > >
> > > > Dear authors,
> > > >
> > > > Thank you for your responses. Since the deadline for submitting recommendation is approaching (Sep 21), can I ask if the revision and your response are finalized and ready for review?

---

> > > > > ### Author Response · Authors · 2024-09-20
> > > > > **Response to Reviewer KcrH**
> > > > >
> > > > > Dear Reviewer,
> > > > >
> > > > > the changes regarding your questions and requests are all contained in our current revision.
> > > > >
> > > > > The only remaining change concerns a request by Reviewer m291, for which we are currently running experiments for a varying number of simulations. First results suggest that the insights shown in our experimental section also hold when the number of simulations is altered.
> > > > > We are planning to include these findings in the coming days as a revision.
> > > > >
> > > > > Please let us know in case there are any additional questions.
> > > > >
> > > > > Best regards,
> > > > >
> > > > > the Authors

---

> > > > > > ### Comment · Reviewer_KcrH · 2024-09-22
> > > > > >
> > > > > > Thank the authors for clearing all my questions and confusions.
> > > > > >
> > > > > > A minor grammatical error:
> > > > > >
> > > > > > "dependencies between agents leads": leads -> lead

---

### Author Response · Authors · 2024-09-06
**Comments on the revision**

Dear Reviewers,

thank you once more for your valuable feedback and engagement!
In the meantime, we have incorporated the textual changes requested by you in the revised version of our paper.

These include:


- More details on the problem statement and the setting of our method and experiments
- Additional information on the challenges to be addressed by our approach
- Further explanations of the benefits brought by topology in the given setting
- More information on the main novelty of the work
- Explanation of the computational complexities of both the Wasserstein distance and the Hausdorff distance, as well as details on the implementation of the Wasserstein distance
- Stating the restriction to low-dimensional parameter spaces as a limitation
- An explanation of Algorithm 2 in sentences (Appendix)
- Fixing the typo in Prop 3.1
- A motivational part for Prop. 3.1 and an interpretation of the statement
- More information in the Introduction on how the given approach fits into the realm on simulation-based inference
- Further details on the Background related to topology and pseudo-marginal MCMC (Appendix)
- More details on what distinguishes this work from other approaches
- Additional motivation for the reason why standard methods are not applicable in our setting
- Further explanations why our experiments fit into the scenario described in the Introduction
- Added MSE loss (as a footnote, p.12) to show that the topological losses of the results shown in Table 2 outperform the geometrical losses
- More details on the model parameters and choices for the experiments

Moreover, a comparison of our results to standard rejection sampling (w.r.t. summary statistics)  is contained in the appendix of our current revision. In the meantime, we are running experiments which showcase results of our experiments for a varying number of simulations, as requested by Reviewer m291. The latter experiments need some time to run, and we are happy to add them later.

Please let us know if there are any other questions or concerns!

Best regards,

the Authors

---

### Author Response · Authors · 2024-09-17
**Update**

Dear Reviewers,

we are currently running the remaining experiments
for a varying number of simulations. First results suggest that
the insights shown in our experimental section also hold when
the number of simulations is altered.

We are planning to include these findings in the coming days
as a revision.

In the meantime, please let us know if there are any questions
or concerns remaining!

Best regards,

The Authors

---

### Decision · Action_Editor_ye22 · 2024-10-06

**Recommendation:** Accept as is

**Comment:**

The authors have already made revisions to the manuscript in response to the reviewer comments and have engaged in a robust discussion with the reviewers about the manuscript. The manuscript is much improved as a result of the revisions to the reviewer comments.

**Audience:**

The reviewers were initially concerned about the appropriate setting for the work and thus the target audience. These concerns were addressed effectively in revision. The paper is focused on settings where the likelihood function is analytically intractable, but for which there is a generative data simulation model to do simulation-based inference such as complex dynamical systems. The reviewers retained concerns about the computational efficiency for high dimensional data, but the idea is sufficiently developed to share widely with a community interested in the intersection of topological data analysis and Bayesian inference.

**Claims And Evidence:**

The paper claims a novel method that bridges the gap between Bayesian methods and topological data analysis methods. The authors support the claim via simulation experiments on lattice-type and percolation models. The reviewers found that the claims largely were supported by the evidence. There were some concerns that the method was not compared to existing baselines. The authors responded that the non-differentiable setting prevented comparison to neural network approaches. The authors did provide a comparison to rejection-sampling methods in the revised version.